# Eurodelta-Trends, a multi-model experiment of air quality hindcast in Europe over 1990-2010.

Colette, Augustin (1), Andersson, Camilla (2), Manders, Astrid (3), Mar, Kathleen (4), Mircea, Mihaela (5), Pay, Maria-Teresa (6), Raffort, Valentin (7), Tsyro, Svetlana (8), Cuvelier, Cornelius (9), Adani, Mario (5), Bessagnet, Bertrand (1), Bergström, Robert (2), Briganti Gino (5), Butler, Tim (4), Cappelletti, Andrea (5), Couvidat, Florian (1), D'Isidoro, Massimo (5), Doumbia, Thierno (10), Fagerli, Hilde (8), Granier, Claire (10,11,12), Heyes, Chris (13), Klimont, Zig (13), Ojha, Narendra (14), Otero, Noelia (4), Schaap, Martin (3), Sindelarova, Katarina (10), Stegehuis, Annemiek I. (15), Roustan, Yelva (7), Vautard, Robert (15), van Meijgaard, Erik (16), Vivanco, Marta Garcia (17), Wind, Peter (8,18),

- (1) INERIS, Institut National de l'Environnement Industriel et des Risques, Verneuil en Halatte, France
- (2) SMHI, Swedish Meteorological and Hydrological Institute Norrköping, Sweden
- (3) TNO, Netherlands Institute for Applied Scientific Research, Utrecht, The Netherlands
- (4) IASS, Institute for Advanced Sustainability Studies, Postdam, Germany
- (5) ENEA - National Agency for New Technologies, Energy and Sustainable Economic Development, Bologna, Italy
- (6) Earth Sciences Department, Barcelona Supercomputing Center-Centro Nacional de Supercomputación, Barcelona, Spain
- (7) CEREA, Joint Laboratory Ecole des Ponts ParisTech - EDF R&D, Champs-Sur-Marne, France
- (8) MET Norway, Norwegian Meteorological Institute, Oslo, Norway
- (9) ex European Commission, Joint Research Centre, Ispra, Italy
- (10) LATMOS/IPSL, UPMC University Paris 06 Sorbonne Universities, Paris, France
- (11) Laboratoire d'Aérologie, Toulouse, France
- (12) NOAA Earth System Research Laboratory and Cooperative Institute for Research in Environmental Sciences, University of Colorado, Boulder, CO, USA
- (13) IIASA International Institute for Applied Systems Analysis, Laxenburg, Austria
- (14) Max-Planck-Institut für Chemie, Mainz, Germany
- (15) LSCE/IPSL, Laboratoire CEA/CNRS/UVSQ, Gif-sur-Yvette, France
- (16) KNMI, Royal Netherlands Meteorological Institute, De Bilt, The Netherlands
- (17) CIEMAT, Madrid, Spain
- (18) Faculty of Science and Technology, University of Tromsø, Tromsø, Norway

## Abstract

The Eurodelta-Trends multi-model chemistry-transport experiment has been designed to facilitate a better understanding of the evolution of air pollution and its drivers for the period 1990-2010 in Europe. The main objective of the experiment is to assess the efficiency of air pollutant emissions mitigation measures in improving regional scale air quality.

The present paper formulates the main scientific questions and policy issues being addressed by the Eurodelta-Trends modelling experiment with an emphasis on how the design and technical features of the modelling experiment answer these questions.

The experiment is designed in three tiers with increasing degree of computational demand in order to facilitate the participation of as many modelling teams as possible. The basic experiment consists of simulations for the years 1990, 2000 and 2010. Sensitivity analysis for the same three years using various combinations of (i) anthropogenic emissions, (ii) chemical boundary conditions and (iii)

meteorology complements it. The most demanding tier consists two complete time series from 1990 to 2010, simulated using either time varying emissions for corresponding years or constant emissions.

Eight chemistry-transport models have contributed with calculation results to at least one experiment tier, and three models have – to date - completed the full set of simulations (and 21-year trend calculations have been performed by four models). The modelling results are publicly available for further use by the scientific community.

The main expected outcomes are (i) an evaluation of the models performances for the three reference years, (ii) an evaluation of the skill of the models in capturing observed air pollution trends for the 1990-2010 time period, (iii) attribution analyses of the respective role of driving factors (emissions/boundary conditions/meteorology), (iv) a dataset  based on a multi-model approach, to provide more robust model results for use in impact studies related to human health, ecosystem and radiative forcing.

# 1    Introduction

Air pollution is a crucial environmental concern because of its detrimental impacts on health, ecosystems, the built environment and short term climate forcing. Whereas it was originally regarded as an urban issue, in the late 1970s the large scale acidification of precipitation made it clear that at least part of the problem could only be solved through international cooperation (OECD, 1977). This was the background for the establishment of the Convention on Long Range Transboundary Air Pollution (CLRTAP) in 1979. The main vehicles of the LRTAP Convention are the Protocols that aim to reduce the emission of various compounds (sulphur in 1985, nitrogen oxides in 1988, volatile organic compounds in 1991, heavy metals and persistent organic pollutants in 1998, and the multi-pollutant multi-effect Gothenburg Protocol to abate acidification, eutrophication and ground level ozone in 1999 and subsequent revision in 2012). The design of such mitigation strategies was largely supported by the development of models (chemistry-transport and integrated assessment tools) and monitoring networks.

After several decades of international cooperation, it is timely to take stock of the evidence available to assess the efficiency of the LRTAP Convention and the corresponding emission ceilings protocols. The Executive Body of the Convention has therefore requested an assessment of the evolution of air pollution and subsequent effects from its two scientific and technical bodies (i) the European Monitoring and Evaluation Programme (EMEP) and (ii) the Working Group on Effects (WGE). As a result, the Task Force on Measurement and Modelling (TFMM) of EMEP published an assessment of air pollution trends (Colette et al., 2016), whereas the WGE published an assessment of corresponding effects on health and ecosystems (De Wit et al., 2015), and an overall assessment report encompassing all the activities undertaken under the Convention was also released (Maas and Grennfelt 2016).

The effects of emissions on the concentrations is rather complex due to (i) the non-linearity of atmospheric chemistry, (ii) the presence of inflow of air pollution due to the intercontinental transport of air pollutants, and (iii) the meteorological variability. This is where Chemistry-Transport Models (CTMs) come into play with the multi-model air quality trend experiment introduced in the present paper.

The LRTAP convention relies in part on the results of the EMEP/MSC-W chemistry-transport model (Simpson et al., 2012a). Since the beginning of the 2000s, the Joint Research Centre of the European Commission initiated a number of multi-model assessments to provide a benchmark for the EMEP/MSC-W model through its comparison with the modelling tools being used by the States-Parties to the Convention as part of the Eurodelta project (Bessagnet et al., 2016;van Loon et al., 2007;Thunis et al., 2008). The Eurodelta-Trends (EDT) exercise builds upon this tradition, focusing on the specific context of air quality trends modelling. Its main goal is to assess to what extent observed air pollution trends could be related to emission mitigation, although this overarching question can only be addressed after having assessed the confidence we can have in the models, in particular in their capacity to reproduce the trends.

Over the recent past, several multi-model projects covering a time period of one year or less were undertaken such as the earlier phases of Eurodelta cited above but also the various phases of the AQMEII project (Galmarini et al., 2012;Galmarini et al., 2017;Rao et al., 2011;Im et al., 2015). However only a few attempts have been made to address the issue of the long-term evolution of European-scale air quality by means of modelling studies. The first attempts were using only one model as in (Vautard et al. (2006);Jonson et al. (2006);Wilson et al. (2012)). A first ensemble was

proposed through the European Project CityZen which relied on 6-models (Colette et al., 2011). While these studies were limited to about 10-yr time periods, a 20-yr hindcast study was presented in (Banzhaf et al., 2015), relying however again on a single model. It is therefore timely to engage in a multi-model hindcast of air quality over two decades.

The purpose of the present paper is to define the science and policy questions that are addressed by the EDT exercise, and introduce the experimental setup designed to answer these questions. The models participating in the experiment will also be presented as well as the project database of model results.

## 2    Experimental design

The main policy focus being addressed in EDT analysis is the assessment of the role of European air pollutant emission reductions in improving air quality over the past two decades. Subsequent questions include assessing (1) the role of changes in global air pollution as well as (2) the role of inter-annual meteorological variability. Before addressing such issues, it will be essential to quantify the CTMs' capability in (1) reproducing observed air pollutant concentrations (processes determining air quality: chemistry, physics, transport processes, emissions, meteorology), and (2) capturing the long term evolution of air quality.

The time period covered by the experiment is 1990-2010. The year 1990 has been chosen as the beginning of the period because that year serves as reference for the Gothenburg protocol. The end of the period is 2010 because of the availability of underlying forcing data (emissions, boundary conditions and meteorology) required for model calculations at the time the work was initiated.

The EDT model experiment is divided into three tiers, targeting various science and policy questions. The tiers also differ in terms of computational demand that allowed involving as many modelling groups as possible. The tiers of experiments are summarized in Table 2. They differ in terms of the number of modelled years to be addressed in the 1990-2010 period and in terms of forcing data used in model calculations for the anthropogenic emissions, the chemical boundary conditions, and the meteorological year. Most of the experiments consist of variations in one or two of these three factors in order to disentangle the role of each forcing. The role of chemical boundary conditions constitutes one notable exception since two sources of forcing are used: either a global CTM simulation or an observation-based climatology (further details are provided on boundary conditions in Section 7).

The first simulation of the EDT experiment is a reference for the year 2010 using the meteorology (M), the chemical boundary conditions (B) and the emissions (E) for that year, named as M10B10E10, with two digits corresponding to the last two digits of the year. They are complemented with simulations for the years 1990 and 2000 (using corresponding meteorology, boundary conditions and emissions: M90B90E90 and M00B00E00 respectively) to form tier 1A. Tier 1A will allow testing the accuracy of all CTMs in simulating pollution changes for the near past (1990, 2000 and 2010), at a lower computational cost than running the full 21-yr period.

Tier 1B is dedicated to the first two sensitivity experiments, for which the meteorology and the boundary conditions are those of the year 2010, but the emissions correspond to 1990 and 2000 (M10B10E90 and M10B10E00). They will allow assessment of the individual impact of emission changes alone (E10 versus E90 and E10 versus E00) by comparison with Tier 1A (specifically M10B10E10).

In Tier 2A, two more sensitivity simulations are performed for the meteorological year 2010, using emissions and boundary conditions of 1990 and 2000 (M10B90E90 and M10B00E00, respectively). By comparison with Tier 1B, they will allow the assessment of the impact of global chemical background changes on European air quality between the years 1990 and 2010, and also for the sub-periods 1990-2000 and 2000-2010 (B10 versus B90 and B10 versus B00).

Tier 2B is an alternate set of reference simulations for 1990, 2000 and 2010, in which boundary conditions provided by a global model (C) instead of the observation-based boundaries (B) are used (M90C90E90, M00C00E00, M10C10E10). It will allow assessment of the uncertainty related to the large scale chemical forcing by comparison with Tier 1A.

Tier 2C is a complement to Tier 2A using the meteorology of 2000 and two combinations of 1990 and 2000 boundary conditions and emissions (M00B90E90, M00B00E90). These additional simulations are required to perform the attribution analysis for the concentration changes between 1990 and 2000, whereas the simulations required for the attribution of driving factors between 1990 and 2010 and between 2000 and 2010 are dealt with in tiers 1A, 1B, and 2A.

Tier 3A consists in 21-year simulations covering 1990-2010, using meteorology, boundary conditions and emissions for the respective years (MyyByyEyy, with yy being the 2-digits year between 1990 and 2010). It will be used to assess the capability of the models to capture observed trends in air quality by means of comparisons with available measurements. Fewer modelling teams delivered results for this higher tier of experiments, therefore model uncertainty will be put in perspective with the spread of the whole ensemble in modelling Tier 1A (1990, 2000, 2010).

Tier 3B is the last sensitivity experiment in which 21-year simulations are performed using the 2010 emissions for the complete period (MyyByyE10). By comparison with Tier 3A, it will allow the determination of the role of inter-annual meteorology and chemical boundary condition changes versus the role of European emission changes.

Thus, the complete series of model runs included for each air quality model is 5 annual simulations for Tier 1, 7 more simulations for Tier 2, and 39 (2x21 minus one overlap for 2010, and two annual simulations belonging to Tier 1A: M90B90E90 and M00B00E00) more simulated years for Tier 3.

Figure 1 provides the schematics of the various combinations of simulations required to perform the attribution analysis for any period of time between the three reference years (1990, 2000 and 2010). The simulations labelled in black are covered by the above simulation plan. They are needed for the assessment of the relative role of emission, meteorology and boundary condition changes.

The main limitations of the simulation plan are (i) that the three selected meteorological years may be not representative, or atypical, for the full period, and (ii) the lack of interaction by considering $2^2$ combinations instead of the $2^3$ combinations required to cover the whole space of factors (Stein and Alpert, 1993). In the forthcoming attribution study these limitations will be explored by (i) comparing trend (tier 3A) and sensitivity (tier 1&2) tiers, and (ii) including additional simulations for the $2^3$ possible combinations from one of the models (Chimere).

## 3    Participating models

Eight European modelling teams submitted their calculation results to the EDT database for at least one tier of experiment (see the experiment design in Section 2) using state-of-art air quality models: Chimere (Menut et al., 2013), CMAQ (Byun and Schere, 2006), EMEP/MSC-W (Simpson et al., 2012), LOTOS-EUROS (Schaap et al., 2008;Manders et al., 2017), MATCH (Robertson et al., 1999) MiNNI

(Mircea et al., 2016), Polyphemus(Mallet et al., 2007), and WRF-Chem (Grell et al., 2005;Mar et al., 2016). The main specifications of the eight participating models are summarized in Table 1 (note that they can differ from the public release of the various models according to the elements provided in the table).

The representation of physical and chemical processes differs in the models. The vertical distribution of model layers (including altitude of the top layer and derivation of surface concentrations at 3m height in the case of EMEP, LOTOS-EUROS and MATCH) is not prescribed either. However, as further explained in the article, the other features of the model setup are largely constrained by the experiment input data such as forcing meteorology, boundary conditions, emissions and by the experiment characteristics such as horizontal domain and resolution. Only one of the participating models included online coupled chemistry/meteorology (WRF-Chem), while all the other models are offline CTMs.

## 4    Modelling domain

The modelling domain is displayed in Figure 2. The domain follows a regular latitude-longitude projection (plate carrée projection) with increments of 0.25° and 0.4° in latitude and longitude, respectively, which is about 25 km x 25 km at European latitudes. The total coverage extends from 17W to 39.8E and from 32N to 70N. All the participating models use the same modelling domain, with only one exception: CMAQB uses a Lambert Conformal Conic projection map with 25 km resolution and delivered their results on the common grid. The south-easternmost part of the domain is not included in the CMAQB modelling domain.

## 5    Meteorology

The horizontal resolution of available global meteorological reanalyses over the 1990-2010 period is considered too coarse to drive regional scale CTMs. Therefore, dynamically downscaled regional climate model simulations using boundary condition from the ERA-Interim global reanalyses (Dee et al., 2011) were used to force the CTMs involved in EDT. Most CTMs used the same meteorological driver, with a couple of exceptions.

One of the meteorological drivers was produced using the Weather Research and Forecast Model (WRF version 3.3.1) (Skamarock et al., 2008) at 0.44 degrees of resolution. In the framework of the EuroCordex climate downscaling programme (Jacob et al., 2013) an evaluation of the regional climate models downscaled with reanalysed boundary conditions (ERA-Interim reanalyses instead of free climate runs) was reported by (Kotlarski et al., 2014). One of the WRF realisations in the EuroCordex ensemble was subsequently further optimized as described in (Stegehuis et al., 2015), so that we could identify an optimal WRF setup for our purpose (row #7 of Table S1 in their supplementary material). The model was re-run using grid nudging towards the ERA-Interim reanalyses (above the planetary boundary layer) in order to improve temporal correlations compared to the regular free-running Cordex hindcast simulations. This WRF simulation was interpolated on the 25km resolution EDT grid and used to drive Chimere, EMEP, Polyphemus, and Minni. In EMEP model, the interpolation of the meteorological fields from 0.4x0.4° to EDT grid was performed online. For WRF-Chem, an online model that simulates meteorology and chemistry simultaneously ("online"), the meteorology from the WRF-Eurocordex runs (Stegehuis et al 2015) was used as initial and lateral boundary conditions and for applying four-dimensional data assimilation (FDDA), with coefficients as described in Mar et al. 2016.The CMAQ model, which runs on a Lambert Conformal Conic projection, could not use the meteorological data provided on the EuroCordex grid, so that WRF was re-run in a

Lambert Conformal projection at 25 km horizontal resolution using identical WRF setup and version (3.3.1).

The CTMs LOTOS-EUROS and MATCH have been meteorologically forced by ERA-Interim series further downscaled with respectively RACMO2 (van Meijgaard, 2012) and HIRLAM (Dahlgren et al., 2016). RACMO2 used here was part of the EuroCordex studies documented in (Jacob et al., 2013) and (Kotlarski et al., 2014) and excludes nudging towards Era-Interim. The HIRLAM EURO4M reanalysis uses data assimilation in 3 dimensions for upper air and optimal interpolation for surface fields. An initial analysis is conducted every 6 hours with subsequent forecasts saved on 3-hourly temporal resolution. ERA-Interim is forced to the lateral boundaries. The HIRLAM reanalysis was interpolated from the original 0.2 horizontal resolution on a rotated lat-lon grid (ca. 22km) to the EDT grid. The main features of the mesoscale meteorological models are synthesized in Table 4.

## 6 Emissions

### 6.1 Annual totals of anthropogenic emissions

National annual emissions, distributed by SNAP (Selected Nomenclature for reporting of Air Pollutants) sectors, were estimated with the GAINS (Greenhouse gases and Air pollution INteractions and Synergies) model (Amann et al., 2011). The calculation was performed for 1990, 1995, 2000, 2005, and 2010 for $SO_2$, $NO_X$, NMVOC, CO, $NH_3$, and PM including $PM_{10}$, $PM_{2.5}$, BC, and OC. To derive emissions for intermediate years, sectorial results for five-year periods were linearly interpolated.

The key activity data originates from Eurostat[1] and International Energy Agency (IEA, 2012) for energy use and from Eurostat, UN Food and Agriculture Organization (FAO)[2], International Fertilizer Association (IFA) for agriculture. For the transport sector, additionally the results of the COPERT model for the EU-28 countries were used (Ntziachristos et al., 2009); this data includes detailed transport sources, fuel distribution, mileage, and level of penetration of control measures. The emission calculation considers impact of existing national and international source specific emission limits and air quality legislation, e.g., several European Union Directives: Large Combustion Plants, Industrial Emissions, National Emission Ceilings Solvent Directive, as well as the UNECE Gothenburg Protocol (UNECE, 1999;Reis et al., 2012). Finally, the results of consultations with national experts, carried within the work on the review of the National Emission Ceiling Directive (Amann et al., 2012) were considered. This emission dataset was completed in April 2014 and is referred to as ECLIPSE_V5; it is part of a global emission set established during the EU funded FP7 project ECLIPSE. More detailed description of the data and applied emission calculation methodology is given in (Amann et al., 2012) and (Klimont et al., 2016b;Klimont et al., 2016a). The respective scenario is available in the freely accessible on-line version of the GAINS model[3] where more detailed outputs and all data inputs can be found.

### 6.2 Spatial distribution of anthropogenic emissions

---

[1] http://ec.europa.eu/eurostat [last access date 14 June 2017]
[2] http://www.fao.org/statistics/en/ [last access date 14 June 2017]
[3] http://magcat.iiasa.ac.at [last access date 14 June 2017]; select 'Europe' in order to access respective data and results

The emissions were provided by INERIS for the EDT modelling domain using the spatial regridding methodology introduced in (Terrenoire et al., 2015;Bessagnet et al., 2016) which consists of:

- Europe-wide road and shipping proxies for SNAP sectors 7 and 8 (road transport and other mobile sources and machinery);
- A proxy based on the population density for residential emissions (SNAP 2: non-industrial combustion plants), note that emissions are not linearly proportional to the population density, a fit tested with the bottom-up inventory for France is used;
- For industrial emissions (SNAP 1, 3, and 4: Combustion in energy and transformation industries; Combustion in manufacturing industry; Production processes) we use the flux and location from the EPRTR inventory[4] . When the total emissions exceed the flux reported in EPRTR, we used a default pattern applying the CEIP spatial distribution, available by SNAP sectors ("emissions as used in EMEP models"[5]). The only exception is for particulate matter emissions for which a spatial distribution was not available for 1990; for that year a combination of officially reported emissions was produced by order of priority: SNAP, NRF01, NFR02 and NFR09 (NFR standing for "Nomenclature for Reporting" following the 2001, 2002, or 2009 guidelines).
- Bottom-up emission inventories for all SNAP for France & United Kingdom (such information was not available elsewhere);
- TNO-MACC inventory for NH3 emissions (largely dominated by SNAP10: agricultural emissions);
- Default CEIP spatial distribution at a 50km resolution for the other sectors (SNAP5, 6, 9: Extraction and distribution of fossil fuels and geothermal energy, Solvents and other product use, Waste treatment and disposal).

In the applied method, only the spatial distribution of industrial emissions is supposed to have changed in time over the past decades. For the residential and road sector, it was considered that the recent techniques involving consistent and high-resolution proxies over Europe provide a more realistic view of emissions than the 50km resolution emission data from the 1990s and early 2000s.

## 6.3 Biogenic and natural emissions

There were no specific constraints imposed to biogenic emissions (including soil NO emission) which are represented by most CTMs using an online module. Forest fires were ignored and each modelling team could decide whether they would include lightning as well as natural and dust emissions from road resuspension of dust emissions (see also the synthesis in Table 1).

## 7 Chemical Boundary Conditions

Two sources of lateral and top chemical boundary conditions are used by the regional CTMs: a climatology of observational data, and global model results. Both have pros and cons. Global models carry biases but include a wider array of chemical species. The trend in observations matches in-situ data by nature, but only at one point over the domain. For the EDT experiment the consensus in the experiment design was in favour of observation-based boundary conditions for most experiments

---

[4] http://prtr.ec.europa.eu [last access date 14 June 2017]
[5] http://www.ceip.at/ [last access date 14 June 2017]

(Tier 1A, 1B, 2A, 2C, 3A, 3B) but also include a sensitivity study based on modelled boundary conditions (Tier 2B).

Note that a possible impact of changing chemistry composition on large scale circulation was integrated in the forcing meteorological fields through the data assimilation of the ERA-Interim reanalysis. This factor was not considered important to isolate for the 2-decade timescale of the experiment.

Note also that both sources are provided on the basis of monthly averages so that sporadic advection of large intercontinental pollution plumes or dust events will not be captured, although their impact on monthly means is taken into account.

## 7.1 Observations-based boundary conditions

The boundary conditions (BCs) are a simplified version of those used in the standard EMEP/MSC-W model (Simpson et al., 2012a). The values are based upon climatological data (except from those for natural particles). The most important gaseous boundary condition compounds are $O_3$, CO and $CH_4$. For ozone, the 3D climatology based on observational vertical profiles constructed by (Logan, 1998) are used in conjunction with a temporal (monthly) variation over the past 20 years. These climatological values are modified each month to ensure that their variability matches the observed variability of concentrations in the clean westerly Atlantic air masses as measured at Mace Head on the coast of Ireland. The 'Mace Head correction' has been derived for each year from ozone data from Mace Head, sorted using sector-analysis (based on trajectories obtained from MSC-W[6]). Monthly mean values of the ozone associated with easterly sectors have been calculated for respective years/months, as described in (Simpson et al., 2012a).

For methane, uniform boundary conditions around the European domain are set to: 1780 ppb in 1990, 1820 ppb in 2000, and 1870 ppb in 2010 according to Mace-Head observations. For the intermediary years, an interpolation is applied.

For sulphate ($SO_4^{2-}$) and nitrate ($NO_3^-$) aerosols, the trends for 1990-2010 are derived from the trend in EPA emissions for North America of $SO_2$ and NOx (Hicks et al., 2002b)[7]. For ammonium ($NH_4^+$), the trends are derived as $2/3*SO_4^{2-} + 1/3*NOx$. The rationale for $SO_2$ lies in the demonstration of the close correspondence between national emissions and concentration trend in (Hicks et al., 2002a).

Monthly (3-dimensional) boundary conditions for sea salt and windblown mineral dust are constructed based on a global run performed with the EMEP/MSC-W model for 2012. The description of EMEP parameterization for sea spray and windblown dust can be found in (Simpson et al., 2012b). The accuracy of the model results for sea salt and mineral dust is regularly evaluated with available observations over Europe and documented in EMEP reports[6]. Model evaluation for mineral dust is limited due to the scarcity of dust in-situ measurements, therefore also AOD/extinction measurements from satellite, Aeronet and Earlinet have recently been used for model evaluation within AeroCom[8].

The uncertainty of these observation-based boundary conditions trends is important and needs to be addressed in the forthcoming analyses of the experiment results, also including a comparison with the model-based boundary conditions.

---

[6] http://www.emep.int [last access date 14 June 2017]
[7] https://www.epa.gov/air-trends [last access date 14 June 2017]
[8] http://aerocom.met.no [last access date 14 June 2017]

## 7.2    Global model based boundary conditions

A global model simulation from the Climate-Chemistry Model Initiative (CCMI) is also used in EDT. CCMI undertakes a global atmospheric chemistry reanalysis over the 1960-2010 time period (Eyring, 2014) based on the MACCity emissions (Granier et al., 2011). The CAM4-chem (Tilmes et al., 2016) member of the CCMI ensemble was made available at monthly temporal resolution for use in EDT.

The model uses a full tropospheric and stratospheric chemistry scheme (Lamarque et al., 2012) based on MOZART (Model for Ozone and Related chemical Tracers) version 4 (Emmons et al., 2010). CAM4-chem considers 56 vertical levels from the surface to about 40 km with 1.9° x 2.5° horizontal resolution. The simulation used in this analysis was performed in nudging the model to meteorological fields from the MERRA GEOS-5 (Modern Era Retrospective Analysis for Research and Application Goddard Earth Observing System Data Assimilation System Version 5) reanalysis provided by the Global Modelling and Assimilation Office (GMAO).

Evaluation of this global reanalysis is ongoing, but the preliminary results are encouraging as illustrated in Figure 3 which shows the modelled and observed ozone trend at the Mace Head station.

## 8    Output format and database status

The model simulations were delivered in a common NetCDF format, so that each of the files contains gridded fields of one pollutant for a whole year. The air pollutant concentrations from only the lowest model level (or corrected to 3m height for EMEP, LOTOS-EUROS and MATCH) are delivered to the project database, but the participants are encouraged to store 3D data if their storage capacities allow such an archiving.

The requested variables are:

- Hourly concentrations of $O_3$ (O3_HL) and $NO_2$ (NO2_HL);
- Daily concentrations of aerosols: nitrate ($NO_3^-$), sulphate ($SO_4^{2-}$) and ammonia ($NH_4^+$), sea-salt, dust, total primary PM, anthropogenic and biogenic secondary organic aerosols, and total PM, both for the fraction below 2.5μm ($PM_{2.5}$), and the fraction below 10μm ($PM_{10}$);
- Daily concentrations of reactive gases: $NH_3$, $SO_2$, an indicator of alpha-pinene that shall depend on the chemical mechanism of each model, isoprene, $HNO_3$, $H_2O_2$, HCHO, PAN, total VOC, biogenic VOC;
- Daily emission rate of biogenic species: isoprene, and an indicator of alpha-pinene that shall depend on the chemical mechanism of each model;
- Monthly dry and wet deposition of total oxidized sulphur (SOx), oxidised nitrogen (NOx) and reduced nitrogen (NHx);
- Hourly meteorological fields: temperature at 2m, wind speed, PBL, rain.

Additional diagnostics were subsequently computed and delivered on the common database, the list of indicators and their definitions is available in Table 5.

The status of models' delivery of results for each of the experiment tiers at the time of submission of the present article is summarized in Table 3. The access to the database is open for research use through the AeroCom server (see also the section on data availability)[9] .

## 9    Sample Results

A few illustrations of the results of the Eurodelta-Trend multi-model air quality hindcast are provided in Figure 4 and Figure 5 with ensemble-median and ensemble-spread concentration maps of ozone and particulate matter in 1990 and 2010, obtained from the eight models which delivered output to Tier 1A. For ozone, we show the summertime (June-July-August) average of the daily maxima of 8-hr mean ozone. For particulate matter, the annual mean $PM_{10}$ is presented.

It is the ambition of the whole Eurodelta-Trend experiment to assess how those maps compare with observations, both in terms of spatial variability and temporal trends, and also to further explain the rationale for the changes observed between 1990 and 2010. Such analyses require however substantial work that is left out of the present article devoted to the presentation of the experiment.

It is worth highlighting however that substantial decreases of both ozone peaks and particulate pollution are modelled in the Eurodelta-Trends ensemble between 1990 and 2010. We present here the decrease on the basis of 1990 and 2010 snapshots for the whole 8 model ensemble that contributed to the experiment. But it would require to be further documented in terms of trends by comparison with the subset of five models that produced the full set of 21-yr trend simulation in tier 3A.

For summertime ozone, concentrations exceeding the European target value of 120 ug/m$^3$ are only found in the greater Mediterranean region in 2010, whereas in the early 1990s, such concentrations affected much larger areas of continental Europe. The spread (standard deviation) of the models is much larger in 1990 than 2010, especially over the polluted areas of Europe at that time.

Particulate matter concentrations also decreased substantially. The largest spread in the eight-member ensemble is found over sea and desert areas (also because absolute concentrations are high over North Africa), where the differences between the models changes significantly between 1990 and 2010. This raises important questions regarding the uncertainties of the models for natural sources and the role of inter-annual meteorological variability on aerosol concentrations.

## 10    Summary and Outlook

The Eurodelta-Trend modelling experiment (EDT) will allow a better understanding of the evolution of regional scale air quality over Europe over the 1990-2010 period. This is facilitated by the thoroughly designed modelling plan. Eight modelling teams have participated in the EDT experiment, though with a variable degree of involvement. The base runs of Tier 1A, completed with eight participating models, offer a great opportunity to assess the capability of these state-of-the-art chemistry-transport models to reproduce the observed changes in the concentrations of the main pollutants, including ozone, particulate matter and its individual components, as well as in precipitation chemistry. This analysis will then be complemented by an assessment of the capability in reproducing the actual trends over the 21yr in the 1990-2010 period for the models participating in the more demanding tier 3A experiment.  If this evaluation phase concludes that the skill of these

---

[9] https://wiki.met.no/aerocom/user-server [last access date 14 June 2017]

models in capturing air quality evolution is satisfactory, we would then rely on the results of the trend (or decadal changes) calculations and the sensitivity experiments and recommend that they can be used when addressing science and policy questions underlying the evolution of air quality in Europe over the past couple of decades.

The critical policy question lies in the attribution of air quality trends to emission changes, to influx at the boundaries of the European domain, and to interannual meteorological variability (and natural sources of trace species) and will be addressed in a series of upcoming papers. Furthermore, thanks to the multi-model design of the experiment, other scientific questions with regard to the role of specific chemical and physical processes will be investigated in forthcoming studies based on the Eurodelta-Trends results.

The model results will also be publicly distributed in order to serve for in depth analyses to scientific communities working on the impacts of air pollution on health, ecosystems or aerosol radiative forcing.

**Data availability**

Technical details allowing forthcoming replication of the experiment are available on the wiki of the EMEP Task Force on Measurement and Modelling[10] and that also provides ESGF links to corresponding input forcing data.

The Eurodelta-Trends model results are made available for public use on the AeroCom server[11] under the following terms:

- Data provided on this server may be used solely for research and education purposes;
- Eurodelta-Trends partners cannot guarantee that the data are correct in all circumstances. Neither do they accept any liability whatsoever for any error or omission in the data, or for any loss or damage arising from its use;
- Data must not be supplied as a whole or in part to any third party without authorization;
- Articles, papers, or written scientific works of any form, based in whole or in part on data, images or other products supplied by Eurodelta-Trends will contain an acknowledgment concerning the supplied data reading:
  - "Modelling data used in the present analysis were produced in the framework of the EuroDelta-Trends Project initiated by the Task Force on Measurement and Modelling of the Convention on Long Range Transboundary Air Pollution. EuroDelta-Trends is coordinated by INERIS and involves modelling teams of BSC, CEREA, CIEMAT, ENEA, IASS, JRC, MET Norway, TNO, SMHI. The views expressed in this study are those of the authors and do not necessarily represent the views of Eurodelta-Trends modelling teams."
- Users of these data must offer co-authorship to the modelling teams for any study submitted for publication until June 2018. The list of modellers is: CHIMERE (A. Colette, F. Couvidat, B. Bessagnet), CMAQ (M.T. Pay), EMEP (S. Tsyro, H Fagerli, P. Wind), ex-JRC (C. Cuvelier), LOTOS-EUROS (A. Manders), MATCH (C. Andersson, R. Bergström), MINNI (M. Mircea, G. Briganti, A. Cappelletti, M. Adani, M. D'Isidoro), POLR (V. Raffort), WRF-Chem (K.A. Mar, N. Otero, N. Ojha). After this date, users must inform the Eurodelta-Trends coordinator

---

[10] https://wiki.met.no/emep/emep-experts/tfmmtrendeurodelta [last access date: 14 June 2017]
[11] https://wiki.met.no/aerocom/user-server [last access date: 14 June 2017]

(augustin.colette@ineris.fr) about the expected use of the data. The coordinator will, in turn, inform a representative from each modelling team.

**Acknowledgements**

- The EMEP MSC-W work has been funded by the EMEP Trust fund and has received support from the Research Council of Norway (Programme for Supercomputing) through CPU time granted at the super computers at NTNU in Trondheim, the University of Tromsø, and the University of Bergen. Michael Schultz and Anna Maria Katarina Benedictow are also gratefully acknowledged for hosting of the Eurodelta-Trends database on the AEROCOM server.
- The GAINS emission trends where produced as part of the FP7 European Research Project ECLIPSE (Evaluating the Climate and Air Quality Impacts of Short-Lived Pollutants); grant no 282688.
- The Chimere simulations where performed were made using the TGCC super computers under the GENCI time allocation gen7485. Also with support from the French Ministry in Charge of Ecology.
- CMAQB simulations were performed on the MareNostrum Supercomputer hosted by the Barcelona Supercomputing Center. The work developed M. T. Pay and its related expenses had been funded by the post-doctoral grant Beatriu de Pinós Program (2011 BP-A2 00015), the CICYT project CGL2013-46736-R and the Severo Ochoa Program awarded by the Spanish Government (SEV-2011-00067).
- Peter Simmonds and Gerry Spain are acknowledged for Mace Head $O_3$ data.
- The CamChem data were produced as part of the CCMI and PEGASOS projects.
- The WRF-Chem simulations have been performed on the supercomputer HYDRA (http://www.rzg.mpg.de/).).
- The computing resources and the related technical support used for MINNI simulations have been provided by CRESCO/ENEAGRID High Performance Computing infrastructure and its staff. The infrastructure is funded by ENEA, the Italian National Agency for New Technologies, Energy and Sustainable Economic Development and by Italian and European research programmes (http://www.cresco.enea.it/english).
- MINNI participation to this project was supported by the "Cooperation Agreement for support to international Conventions, Protocols and related negotiations on air pollution issues", funded by the Italian Ministry for Environment and Territory and Sea.RACMO2 simulations at KNMI to provide meteorological forcings for LOTOS-EUROS were supported by the Dutch Ministry of Infrastructure and the Environment.
- The MATCH participation was partly funded by the Swedish Environmental Protection Agency through the research program Swedish Clean Air and Climate (SCAC) and NordForsk through the research programme Nordic WelfAir (grant no. 75007).
- RACMO2 simulations at KNMI to provide meteorological forcings for LOTOS-EUROS were supported by the Dutch Ministry of Infrastructure and the Environment.

1    Table 1: Main features of the Chemistry-Transport Models involved in the Eurodelta-Trends modelling exercise

| MODEL | CHIMERE | CMAQB | EMEP | LOTOS-EUROS | MATCH | MINNI | POLYPHEMUS | WRF-CHEM |
|---|---|---|---|---|---|---|---|---|
| **version** | Modified Chimere2013 | V5.0.2 | rv4.7 | v1.10.005 | VSOA April 2016 | V4.7 | V1.9.1 | V3.5.1 |
| **operator** | INERIS | BSC | MET Norway | TNO | SMHI | ENEA/Arianet S.r.l. | CEREA | IASS |
| **Chemistry/Meteorology coupling** | offline | offline | offline | offline | offline | offline | Offline | Online |
| **Name and resolution of the meteorological driver** | WRF (common driver after (Stegehuis et al., 2015)). 0.44 deg. | WRF. 25km | WRF (common driver after (Stegehuis et al., 2015)). 0.44 deg. | RACOMO2, 0.22 deg. | HIRLAM EURO4M reanalysis, approx. 22km | WRF (common driver after (Stegehuis et al., 2015)). 0.44 deg. | WRF (common driver after (Stegehuis et al., 2015)). 0.44 deg. | WRF, approx. 25km (common driver used for initial and lateral boundary conditions, and for applying four-dimensional data assimilation (FDDA), with coefficients as described in (Mar et al. 2016). |
| **Vertical layers** | 9 sigma | 15 sigma | 20 sigma | 5(4 dynamic layers and a surface layer) | 39 hybrid eta utilizing the meteorological model layers | 16 fixed terrain-following layers | 9 Fixed terrain following layers | 35 terrain-following |
| **Vertical extent** | 500 hPa | 50 hPa | 100 hPa | 5000 m | ca 5000 m (4700 – 6000 m) | 10000 m | 12000m | 10 hPa |
| **Depth first layer** | 20 m | 40 m | 90 m | 25 m | ca 60m | 40 m | 40m | 50 m |
| **Surface concentration** | First model level | First model level | Downscaled to 3m using dry deposition velocity and similarity theory | Downscaled to 3m | Downscaled to 3m | First model level | First model level | First model level |
| **Biogenic VOC** | MEGAN model v2.1 with high resolution spatial and temporal LAI (Yuan et al., 2011) and recomputed emissions factors | MEGAN model v2.04 (Guenther et al., 2006) | Based upon maps of 115 species from (Koeble and Seufert, 2001), and hourly temperature and light using (Guenther et al., | Based upon maps of 115 species from (Koeble and Seufert, 2001), and hourly temperature and light (Guenther et al 1991, | (Simpson et al., 2012a), based on hourly temperature and light. | MEGAN v2.04 (Guenther et al., 2006) | MEGAN V2.04 (Guenther et al., 2006) | MEGAN v2.04 (Guenther et al., 2006) |

| | | | | | | | | |
|---|---|---|---|---|---|---|---|---|
| | based on the landuse (Guenther et al., 2006) | | 1993;Guenther et al., 1994). See (Simpson et al., 1995;Simpson et al., 2012a) | Guenther et al 1993). See (Beltman et al., 2013) | | | | |
| **Forest fires** | None | None | None | None | None | None | None | None |
| **Soil-NO** | MEGAN model v2.04 | MEGAN model v2.04 | See in (Simpson et al., 2012a) | Not used here | None | MEGAN v2.04 | MEGAN V2.04 | MEGAN v2.04 |
| **Lightning** | None | None | Monthly climatological fields, (Köhler et al., 1995) | None | None | None | None | None |
| **Sea salt** | (Monahan, 1986) | Open ocean and surface-zone (Kelly et al., 2010) | (Monahan, 1986) and (Martensson et al., 2003), see (Tsyro et al., 2011) | (Martensson et al., 2003) and (Monahan, 1986), see (Schaap et al., 2009) | Based on parameterization by (Sofiev et al., 2011) | (Zhang K.M. et al., 2005) | (Monahan, 1986) | (Gong et al., 1997), (O'Dowd et al., 1997) |
| **Windblown Dust** | (Vautard et al., 2005), not used here | None | See (Simpson et al., 2012a) | (Schaap et al., 2009) | Not used here | (Vautard et al., 2005) | None | None |
| **Dust traffic suspension** | None | None | (Denier van der Gon et al., 2010) | None | Not used here | None | None | None |
| **Landuse database** | GLOBCOVER (24 classes) | Corine Land Cover 2006 (44 classes) | CCE/SEI for Europe, elsewhere GLC2000 | Corine Land Cover 2000 (13 classes) | CCE/SEI for Europe | Corine Land Cover 2006 (22 classes) | Global Land Cover 2000 (24 classes) | 24-category USGS landuse |
| **Advection scheme** | (van Leer, 1984) | Horizontal: WRF-based scheme, vertical: Piecewise Parabolic Method | (Bott, 1989) | (Walcek, 2000) | Fourth order mass-conserved advection scheme based on (Bott, 1989) | Blackman cubic polynomials (Yamartino, 1993) | Third-order Direct Space Time scheme (Spee, 1998) with Koren-Sweby flux limiter function | Runge-Kutta 3rd order |
| **Vertical diffusion** | Kz approach following (Troen and Mahrt, 1986) | ACM2 PBL scheme (Pleim, 2007) | Kz approach following (O'Brien, 1970) and (Jeričevič et al., 2010) | Kz approach Yamartino et al (2004) | Implicit mass conservative Kz approach, see (Robertson et al., 1999)<br><br>Boundary layer parameterisation as detailed in (Robertson | Kz approach following (Lange, 1989) | Kz approach following (Troen and Mahrt, 1986) | Yonsei University PBL scheme (Hong et al., 2004) |

| | | | | | | | | |
|---|---|---|---|---|---|---|---|---|
| | | | | | et al., 1999) forms the basis for vertical diffusion and dry deposition | | | |
| **Dry deposition** | Resistance approach (Emberson et al., 2000a;Emberson et al., 2000b) | Resistance approach (Venkatram and Pleim, 1999) | Resistance approach for gases (Venkatram and Pleim, 1999) for aerosols, (Simpson et al., 2012a) | Resistance approach,DEPAC3.11 for gases, (Van Zanten et al., 2010) and (Zhang et al., 2001) for aerosols | Resistance approach depending on aerodynamic resistance, and land use (vegetation). Similar to (Andersson et al., 2007) | Resistance model based on (Wesely, 1989) | Resistance approach for gases (Zhang et al., 2003) and aerosols (Zhang et al., 2001) | (Wesely, 1989) and (Erisman et al., 1994) |
| **Ammonia compensation points** | None | None | None, but zero $NH_3$ deposition over growing crops | Only for $NH_3$ (for stomatal, external leaf surface and soil (= 0)) | None | None | None | None |
| **Stomatal resistance** | (Emberson et al., 2000a;Emberson et al., 2000b) | (Wesely, 1989) | DO3SE-EMEP: (Emberson et al., 2000a;Emberson et al., 2000b), (Tuovinen et al., 2004;Simpson et al., 2012a) | (Emberson et al., 2000a;Emberson et al., 2000b) | Simple, seasonally varying, diurnal variation of surface resistance for gases with stomatal resistance (similar to (Andersson et al., 2007)) | (Wesely, 1989) | (Zhang et al., 2003) | (Wesely, 1989) and (Erisman et al., 1994) |
| **Wet deposition gases** | In-cloud and sub-cloud scavenging coefficients | In-cloud and sub-cloud scavenging which depends on Henry's law constants, dissociation constants and cloud water pH (Chang et al., 1987) | In-cloud and sub-cloud scavenging coefficients | sub-cloud scavenging coefficient | In-cloud scavenging of some species based on Henry's law constants. Simple in-cloud and sub-cloud scavenging coefficients for other gases. | In-cloud and sub-cloud scavenging coefficients (EMEP, 2003) | In-cloud (monodispersed raindrops with constant collection efficiency) and bellow cloud (Sportisse and Dubois, 2002) scavenging coefficients | In-cloud and sub-cloud scavenging coefficients |
| **Wet deposition particles** | In-cloud and sub-cloud scavenging | In-cloud and sub-cloud scavenging | In-cloud and sub-cloud scavenging | sub-cloud scavenging coefficient | In-cloud and sub-cloud scavenging. Similar to (Simpson et al., 2012a) | In-cloud and sub-cloud scavenging coefficients | In-cloud (as for gas) and bellow cloud (Slinn, 1983) scavenging coefficients | In-cloud and sub-cloud scavenging coefficients |
| **Gas phase chemistry** | MELCHIOR2 | CB-05 with chlorine chemistry extensions | EmChem09 (Simpson | TNO-CBM-IV | Based on EMEP (Simpson et al., 2012), | SAPRC99 | CB-05 (Yarwood. G. et | RADM2 (Stockwell et al., 1990) with |

| | | | | | | | | |
|---|---|---|---|---|---|---|---|---|
| | | (Yarwood. G. et al., 2005) | et al., 2012a) | | with modified isoprene chemistry (Carter, 1996;Langner et al., 1998) | (Carter;Carter, 2000) | al., 2005) | updates made to inorganic rate coefficients as described in supplementary material to (Mar et al. 2016). |
| **Cloud chemistry** | Aqueous $SO_2$ chemistry and ph dependent $SO_2$ chemistry | Aqueous $SO_2$ chemistry (Walcek and Taylor, 1986) | Aqueous $SO_2$ chemistry, pH dependent | Aqueous SO2 chemistry, pH dependent (Banzhaf et al., 2012) | Aqueous $SO_2$ chemistry | Aqueous $SO_2$ chemistry (Seinfeld and Pandis, 1998) | Aqueous $SO_2$ chemistry (Seinfeld and Pandis, 1998) | None |
| **Coarse nitrate** | No reaction with Ca even if reaction with Na is taken into account. Coarse nitrate might exists with transfer from smaller particles | None | Two formation rates of coarse $NO_3$ from $HNO_3$ for relative humidity below/above 90% | (Wichink Kruit et al., 2012) | Yes, transfer of $HNO_3(g)$ to aerosol nitrate using rate from (Strand and Hov, 1994) | None | No heterogeneous nitrate formation | None |
| **Ammonium nitrate equilibrium** | ISORROPIA v2.1 (Nenes et al., 1999) | ISORROPIAv2.1 | MARS (Binkowski and Shankar, 1995) | ISORROPIA v.2 | RH & T dependent equilibrium constant (Mozurkewich, 1993) | ISORROPIA v1.7 (Nenes et al., 1998) | ISORROPIA v1.7 (Nenes et al., 1999) | MARS (Binkowski and Shankar, 1995) |
| **SOA formation** | $H_2O$ (Couvidat et al., 2012) mechanism coupled with the thermodynamic model SOAP (Couvidat and Sartelet, 2015) | SORGAM module (Schell et al., 2001) | VBS-NPAS –(Simpson et al., 2012a) | Not used here | Similar to VBS-NPNA (Bergström et al., 2012) | SORGAM module (Schell et al., 2001) | $H_2O$ (Couvidat et al., 2012) | SORGAM module (Schell et al., 2001) |
| **Volatility basis set for aerosols** | None | None | (Simpson et al., 2012a;Bergström et al., 2012) | Not used here | Yes, based on Bergström et al. (2012) | None | None | None |
| **Aerosol model** | 9 bins (10 nm to 10 µm) | AERO5 (Carlton et al., 2010), log-normal approach (3 modes) | Bulk- approach (fine and coarse modes) | Bulk- approach (2 modes) | Bulk approach | AERO3 (Binkowski, 1999); 3 modes: Aitken, accumulation, coarse | 5 bins (0.01 to 10µm) | MADE (Ackermann et al., 1998) |
| **Aerosol physics** | coagulation/condensation/nucleation | Coagulation/condensation/nucleation | Not used here | Not used here | Not used here | Coagulation/condensation/nucleation | Coagulation/Condensation | Coagulation/condensation/nucleation |

| | Computation of the wet diameter for each bins as a function of humidity (used for coagulation, condensation, deposition) | | | | | | |
|---|---|---|---|---|---|---|---|

1     Table 2 : Summary of model experiments (including label), corresponding key scientific questions.
2     The simulations are labelled MyyByyEyy where M indicates meteorology, B indicates observation-
3     based boundary conditions, C indicates modelled-based boundary condition, E indicates emission,
4     and yy is the 2-digits reference of the corresponding year.

| Tier | Experiment | Key question (Q) / Action (A) | Label |
|------|-----------|-------------------------------|-------|
| 1A | Meteorology,boundary conditions and emissions of 1990, 2000 and 2010. | Q: What is the uncertainty within the seven CTMs ensemble in 1990, 2000, and 2010? A: Comparison 1A vs. Observations for 1990, 2000 and 2010 | M10B10E10 M00B00E00 M90B90E90 |
| 1B | Meteorology and boundary conditions of 2010. Emissions of 1990 and 2000. | Q: What if no emission change occurred in Europe? A: Comparison 1A vs. 1B | M10B10E00 M10B10E90 |
| 2A | Meteorology of 2010. Emissions and boundary conditions of 1990 and 2000. | Q: What if no emission changed beyond Europe? A: Comparison 2A vs. 1B | M10B00E00 M10B90E90 |
| 2B | Meteorology and emissions of 2010. Modelled boundary conditions of 1990, 2000, 2010 | Q: What is the uncertainty related to boundary conditions? A: Comparison 2A vs. 2B | M10C10E10 M00C00E00 M90C90E90 |
| 2C | Meteorology of 2000, emissions of 1990 and boundary conditions of 2000 and 1990. | Additional simulations for decomposition of factors in the 1990s and 2000s | M00B90E90 M00B00E90 |
| 3A | 21-years reference trend from 1990 to 2010 | Q: How do the models capture the trend in observations? A: Comparison 3A vs. observations | MyyByyEyy |
| 3B | 21-years trend with 2010 emissions | Q: Does meteorological variability contribute to the AQ trend over the past 20 years? A: Comparison 3A vs. 3B | MyyByyE10 |

Table 3: Synthesis of models having delivered (D) data or planning to (P) to the project database for each of the experiments.

| Tier | Label | CHIMERE | CMAQB | EMEP | LOTOS-EUROS | MATCH | MINNI | Polyphemus | WRF-Chem |
|---|---|---|---|---|---|---|---|---|---|
| 1A | M10B10E10 | D | D | D | D | D | D | D | D |
|  | M00B00E00 | D | D | D | D | D | D | D | D |
|  | M90B90E90 | D | D | D | D | D | D | D | D |
| 1B | M10B10E00 | D | D | D | D | D | D | D | D |
|  | M10B10E90 | D | D | D | D | D | D | D | D |
| 2A | M10B00E00 | D | D | D | D | D | D | D | P |
|  | M10B90E90 | D | D | D | D | D | D | D | P |
| 2B | M10C10E10 | D |  | D |  |  | D |  | P |
|  | M00C00E00 | D |  | D |  |  | D |  | P |
|  | M90C90E90 | D |  | D |  |  | D |  | P |
| 2C | M00B90E90 | D | D | D | D | D | D | D | P |
|  | M00B00E90 | D | D | D | D | D | D | D | P |
| 3A | MyyByyEyy | D |  | D | D | D | D |  | P |
| 3B | MyyByyE10 | D |  | D | D | D | D |  |  |

Table 4: Meteorological fields used in the EDT project. WRF-0.44 corresponds an optimized and nudged version of the WRF-IPSL-INERIS Eurocordex member at 0.44 degrees from EuroCordex climate downscaling programme (Jacob et al., 2013) used by most CTMs in EDT. WRF-25 corresponds to the WRF run in the same condition as WRF-0.44 in a Lambert Conformal Conic projection used to drive CMAQB. WRF-Chem indicates the configuration of WRF used within the WRF-Chem online CTM. RACMO2 is the meteorological model used by LOTOS-EUROS.

| Model configuration | WRF-0.44 | WRF-25 | WRF-Chem | HIRLAM EURO4M | RACMO2 |
|---|---|---|---|---|---|
| Model version | WRF v3.3.1 | WRF v3.3.1 | WRF v3.5.1 | HIRLAM 3DVAR upper air analysis and OI surface analysis (for details and evaluation see (Dahlgren et al., 2016) | RACMO2.3 (Meijgaard et al., 2012) |
| Initial and boundary conditions | ERA-Interim global reanalysis (resolution ~80 km) (Dee et al., 2011) | ERA-Interim global reanalysis (resolution ~80 km) (Dee et al., 2011) | WRF-0.44 simulation used by other EDT models | ERA-Interim global reanalysis (resolution ~80km) (Dee et al., 2011) | ERA-Interim global reanalysis (resolution ~80 km) (Dee et al., 2011) |
| Coordinate system | Rotated latitude and longitude | Lambert Conformal | Latitude and longitude | Rotated latitude and longitude | Rotated latitude and longitude with a South Pole at 47S and 10E. |
| Horizontal setting / number of zonal and meridional grid cells | 0.44º x 0.44º (120-117) | 25 km x 25 km (176-197) | Approx. 25 km x 25 km (144-154) | Approx. 22km x 22km (326-341) | 0.22x0.22 (306x220) |
| Vertical setting | 31 layers | 31 layers | 34 layers | 60 layers eta coordinates | 40 layers hybrid coordinates |
| Microphysics | Morrison DM (Morrison et al., 2009) | Morrison DM (Morrison et al., 2009) | Morrison DM (Morrison et al., 2009) | Large-scale condensation with Rasch-Kristjansson scheme (Rasch and Kristjánsson, 1998) | Prognostic cloud scheme (Tiedtke, 1993), Large-scale condensation (Tompkins et al., 2007), boundary-layer clouds (Neggers, 2009) |
| LW,RW radiation | RRTMG - (Iacono et al., 2008) | RRTMG - (Iacono et al., 2008) | RRTMG - (Iacono et al., 2008) | (Savijärvi, 1990) | Short wave radiation (Clough et al., 2005;Morcrette et al., 2008) Long wave radiation (Mlawer et al., 1997;Morcrette et al., 2001) |

| | | | | | |
|---|---|---|---|---|---|
| **Cumulus scheme** | Tiedtke - (Tiedtke, 1989;Zhang et al., 2011) | Tiedtke - (Tiedtke, 1989;Zhang et al., 2011) | Grell 3D scheme[12] (Grell and Dévényi, 2002) | Convective processes Kain-Fritsch scheme (Kain, 2004) | Mass flux scheme (Tiedtke, 1989;Nordeng, 1994;Neggers et al., 2009;Siebesma et al., 2007) |
| **Boundary & Surface layer** | MYNN-ETA (Janjic, 2002;Nakanishi and Niino, 2006;Nakanishi and Niino, 2009) | MYNN-ETA (Janjic, 2002;Nakanishi and Niino, 2006;Nakanishi and Niino, 2009) | MYNN-ETA (Janjic, 2002;Nakanishi and Niino, 2006;Nakanishi and Niino, 2009) | Turbulence CBR scheme (Cuxart J. et al., 2000); adaptions for moist CBR (Tijm and Lenderink, 2003) | Eddy-Diffusivity Mass Flux Scheme with TKE prognostic variable (Lenderink and Holtslag, 2004;Siebesma et al., 2007) |
| **Soil** | NOAH (Tewari et al., 2004) | NOAH (Tewari et al., 2004) | NOAH (Tewari et al., 2004) | Further developed ISBA scheme (Noilhan and Planton, 1989;Noilhan J. and J.-F., 1996;Gollvik and Samuelsson, 2010) | TESSEL (Van den Hurk et al., 2000), HTESSEL (Balsamo et al., 2009) |

---

<sup>12</sup> A different scheme was chosen for compatibility with chemistry, in particular so that there would be subgrid convective transport of chemical species.

| | |
|---|---|
| O3_DL | daily ozone computed on the basis O3_HL as the mean value for each day between 00:00 and 23:00 UTC. |
| O3Aot40_DL | accumulated ozone over 40ppbv computed on the basis of O3_HL, for each day (from 1 May until 31 July) as the sum of all the daylight hourly O3_HL values exceeding the value of 40 ppb (80 ug/m3). Note that hourly values in the models correspond to instantaneous values: e.g. O3_HL(0) is for 00:00, O3_HL(23) is for 23:00). Therefore, the accumulation of AOT between 8hr and 20hr was taken as the sum of O3_HL between 8:00 and 19:00, included. O3Aot40_DL is a daily quantity that must be cumulated over a given period of the year, e.g. May-June-July in the European Air Quality Directive (EC, 2008). Its units are: $(\mu g/m^3)$*hours |
| O3Aot60_DL | same as before, but with a threshold of 60 ppb (120 $\mu g/m^3$) and to be accumulated over the period 1 April - 30 September. Its units are: $(\mu g/m^3)$*hours |
| O3hr8_HL | the 8-hr running mean hourly ozone computed from O3_HL. To each hour ih in O3hr8_HL the running mean is that of the 8 past values of O3_HL : O3hr8_HL(ih) = [ O3_HL(ih) + ... O3_HL(ih-7) ] /8 |
| O3hr8Somo35_DL | Sum of ozone means over 35ppbv computed from O3hr8_HL for each day of the year as the exceedance of the daily max O3hr8_HL with respect to 35 ppb (70 $\mu g/m^3$). The accumulated value used in the Air Quality Directive is the sum over all days of the year. Its units are: $(\mu g/m^3)$*days |
| O3hr8Max_DL | Maximum daily value of O3hr8_HL, sometimes also referred to as MDA8 as Ozone Maximum Daily Average |
| O3hr8Exc60_DL | computed from O3hr8_HL. For each day of the year a value of 1 is assigned if the maximum daily value of O3hr8_HL exceeds 60 ppb (120 $\mu g/m^3$), otherwise equal to zero. The value mentioned in the Directive is the sum over all days of the year. Units are: days. |
| NO2_DL | Computed from NO2_HL, same as O3_DL |
| NO2hr1Max_DL | Computed from NO2_HL, Maximum daily value of NO2_HL |
| NO2hr1Exc200_DL | Computed from NO2_HL, for each day of the year a value of 1 is assigned if the maximum daily value of NO2_HL exceeds 200 ppb, otherwise equal to zero. The value mentioned in the Directive value is the sum over all days of the year. Its units are: days |
| NOx-ppb DL and HL | Sum of NO and NO2 in ppb, i.e. NO($\mu g/m^3$)*22.4/30 + $NO_2$($\mu g/m^3$)*22.4/46 |
| PM10Exc50_DL | Computed from daily mean $PM_{10}$ (PM10_DL), for each day of the year a value of 1 is assigned if the (daily) value of PM10_DL exceeds 50 ug/m3, otherwise equal to zero. The value in the Directive is the sum over all days of the year. Units are: days |
| TNO3-N | Sum of NO3-10 and HNO3 in $\mu gN/m^3$, i.e. NO3-10($\mu g/m^3$)*14/62 + HNO3($\mu g/m^3$)*14/63 |
| TNH4-N | Sum of NH4-10 and NH3 in $\mu gN/m^3$: i.e. NH4-10($\mu g/m^3$)*14/18 + NH3($\mu g/m^3$)*14/17 |
| TSO4-S | Sum of SO4-10 and SO2 in $\mu gS/m^3$: SO4-10($\mu g/m^3$)*32/96 + SO2($\mu g/m^3$)*32/64 |
| NOz | Sum of HNO3, and PAN in ppb. Conversion factors from $\mu g/m^3$ to ppb: [24/63,24/53] |
| NOy | Sum of NO2, NO, HNO3, and PAN in ppb. Conversion factors from $\mu g/m^3$ to ppb: [24/46,24/30,24/63,24/53] |

Table 5: List and definition of air pollution indicators derived from the model results and available in the project database

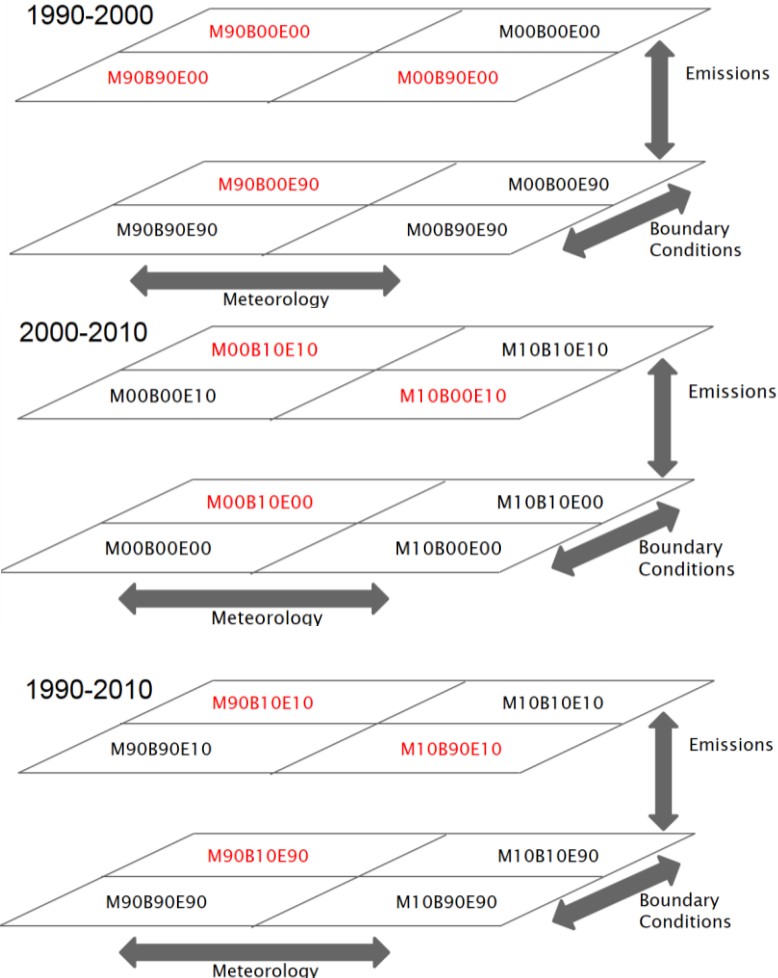

Figure 1 : Combination of sensitivity simulations required to perform the analysis of the contribution of (i) meteorology, (ii) boundary conditions, and (iii) emission changes for the 1990-2000, 2000-2010, and 1990-2010 years from the top to the bottom. The key to EDT model simulations provides the 2-digit modelled year for meteorology (M), boundary conditions (B) and emissions (E). Black labels are for the simulations included in the experiment, and red labels are the combinations not produced in any of the tiers of experiments.

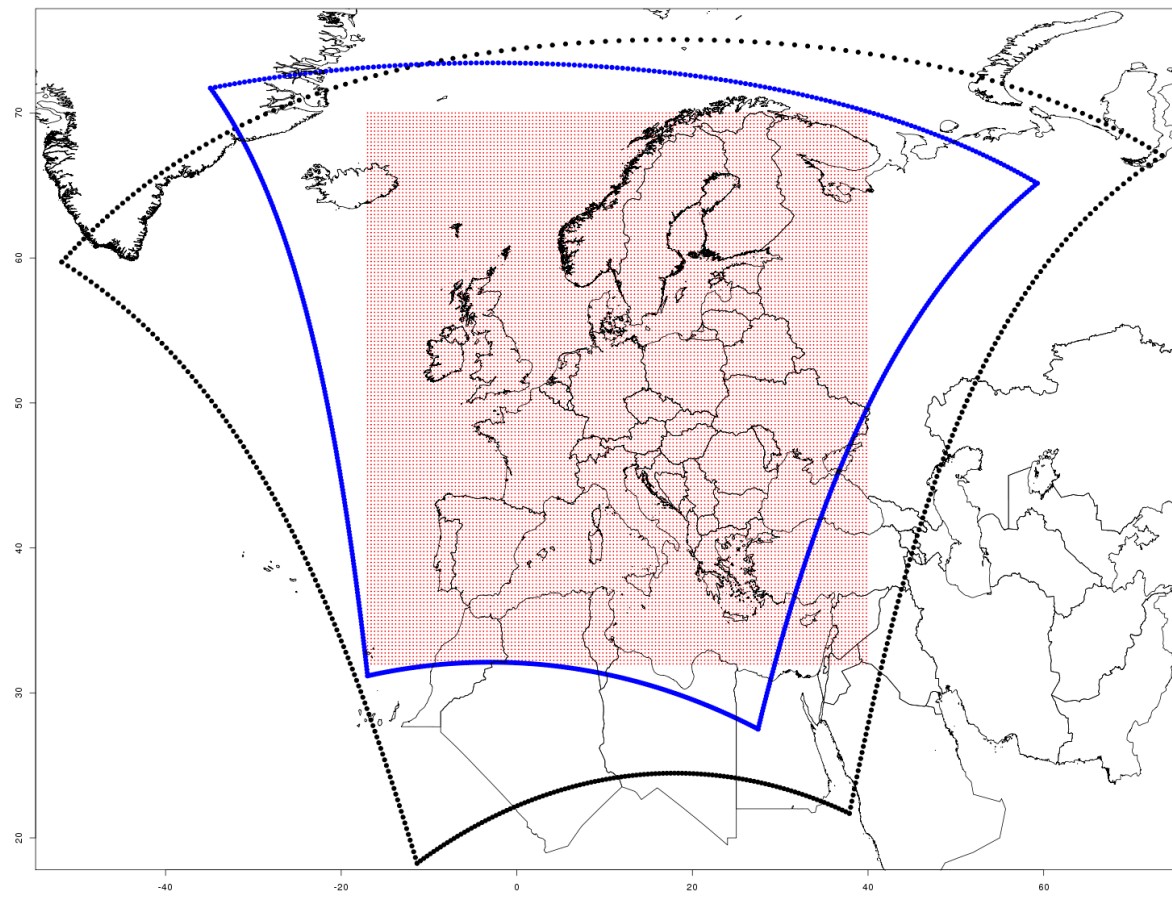

2  Figure 2: Modelling grid used by all the chemistry transport models involved in Eurodelta-Trends (red dots)
3  with the exception of CMAQB that could not implement a regular latitude/longitude grid (outer grid cell of the
4  modelling domain displayed with blue dots). The outer grid cells of the meteorological forcing data on the
5  EuroCordex grid is also displayed (black dots).

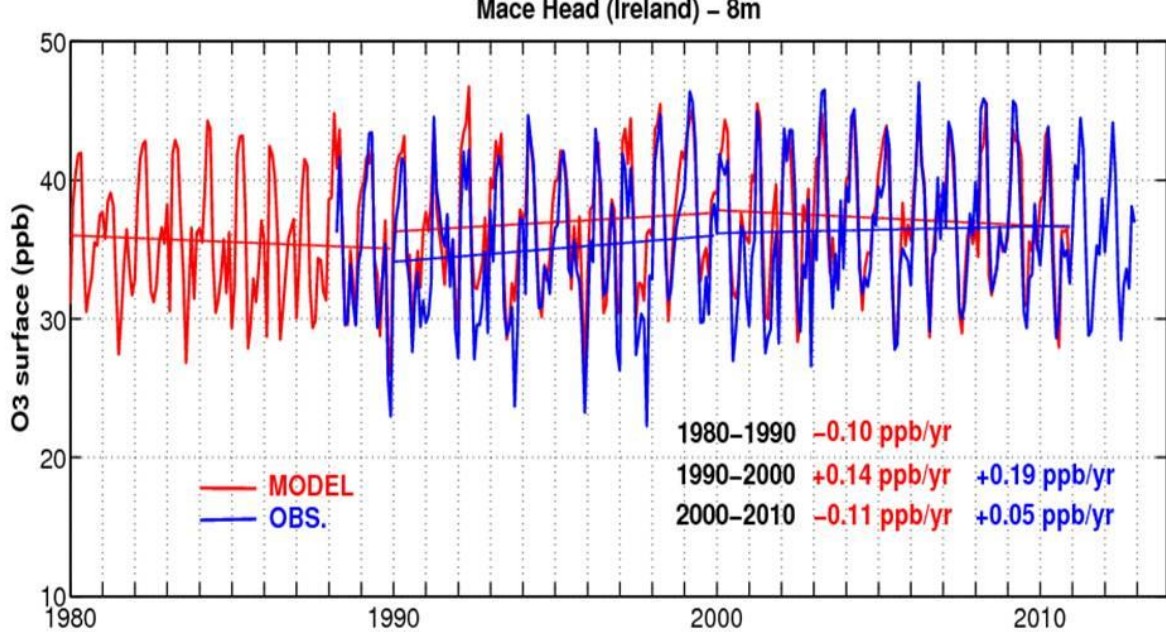

3    Figure 3 : Monthly variation of surface ozone (in ppb/year) at the Mace Head station observed (blue)
4    and modelled (red) in the CamChem member of the Climate-Chemistry Model Initiative (CCMI)

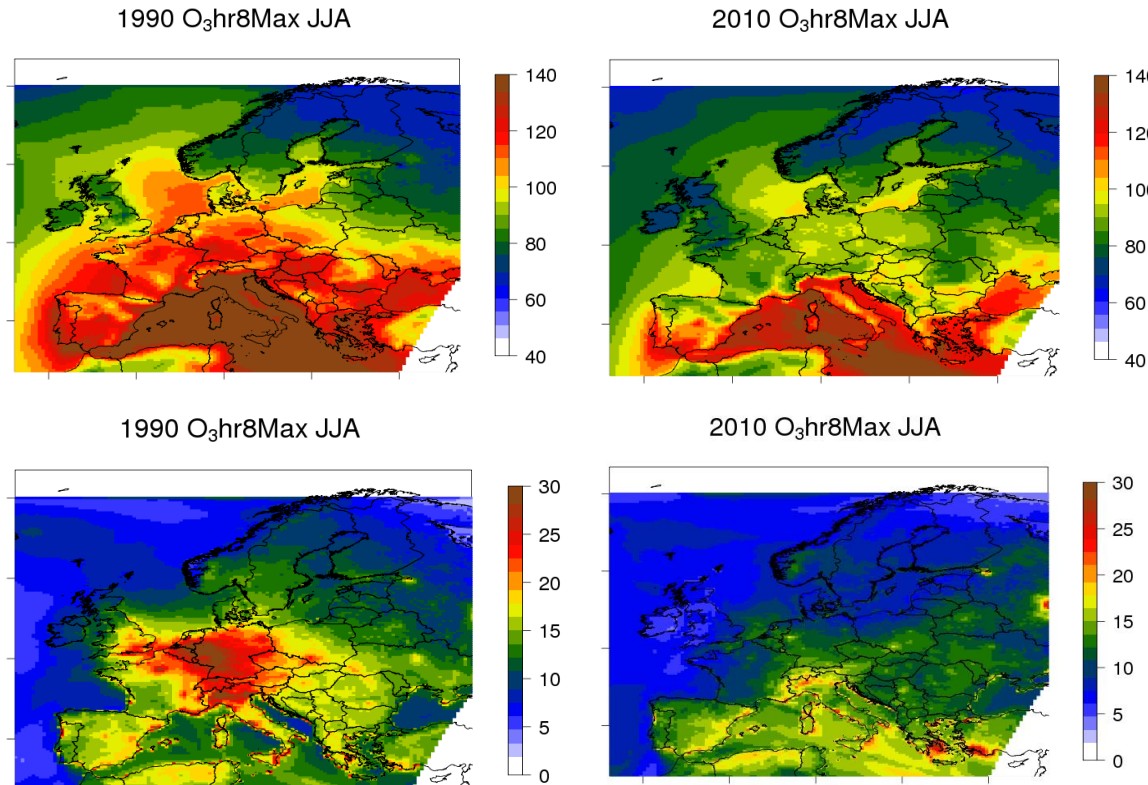

1  Figure 4: Eight-model ensemble results for 1990 (first column) and 2010 (second column) for
2  summertime ozone peaks (June-July-August means of 8-hr mean daily maxima, μg/m³). Top:
3  ensemble median, bottom: ensemble spread (standard deviation).

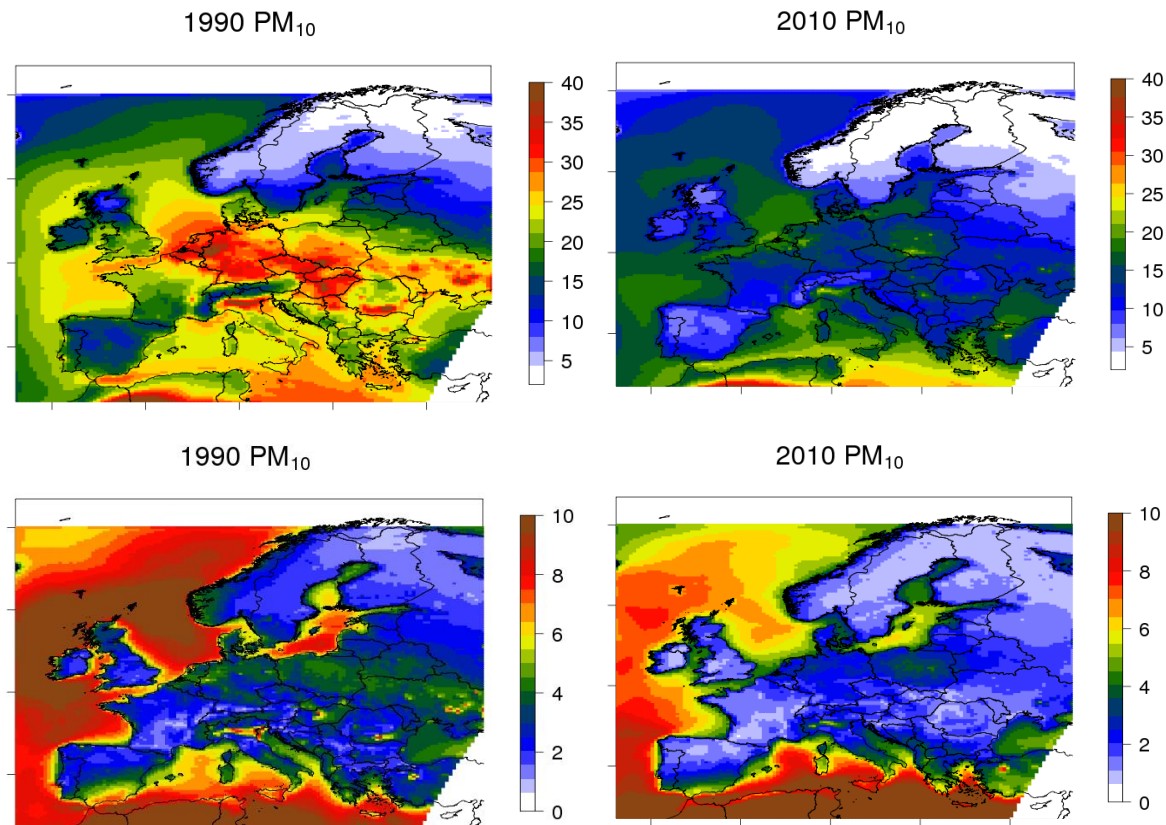

Figure 5: Eight-models ensemble results for 1990 (first column) and 2010 (second column) for annual mean $PM_{10}$ ($\mu g/m^3$). Top: ensemble median, bottom: ensemble spread (standard deviation).

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
