# Peer review of "Eurodelta-Trends, a multi-model experiment of air quality hindcast in Europe over 1990-2010."

_Geoscientific Model Development, 2016_

## Referee Comment (RC1)

**Title: EURODELTA-Trends, a multi-model experiment of air quality 1 hindcast in Europe over 1990-2010.**

**General comment:**
The manuscript provides detailed information of the experiment set up. The experiments look reasonable. However, it is very difficult to understand the anticipated results based on the current information. As described in the journal website, results should be provided for model experiment description papers.
([http://www.geoscientific-model-development.net/about/manuscript_types.html#item4](http://www.geoscientific-model-development.net/about/manuscript_types.html#item4))

*"Should include sufficient descriptions/figures of model results to give an overview of the project."*

However, the manuscript does not provide any results. Without results, it is not easy for referees to provide further meaningful comments on the experiments.

**Specific comments:**
Line 12 in page 3: the "(" should be removed.

Line 39-40 in page 3: the sentence is not completed.

Line 32 in page 6: what does the "perfect" mean? I assume the authors refer to "better".

Line 34-35 in page 8: What is the reason to rely on observation-based boundary for most experiments?

Line 18 in page 9: The ammonium trend based on $NO_x$ is not necessary true, as $NH_4$ is also affected by $NH_3$ emission which is expected to maintain at a constant level in the future, while $NO_x$ is expected to reduce.

Line 34 in page 9: "The air concentration" should be "The air pollutant concentration".

Line 18 in page 12: The referee cannot agree this requirement based on two reasons. First, if users use the data, they should cite the corresponding publications. This should provide enough credits for the authors. Second, the findings of other studies may not be agreed by the authors. This may not be appropriate to include the authors' names on a paper that has findings against the authors' understanding. Therefore, this requirement should be modified such as "users of these data should cite this paper".

---

## Referee Comment (RC2) · Anonymous Referee #2 · 9 Mar 2017

The paper describes a multi-model experiment including 12 one year long air quality simulations for Europe and two simulations with a length of twenty years. The volume of work presented in this model experiment description paper is really impressive and the paper should be published after some revision as a basis for the hopefully following numerous papers going more into the details.

In its currents form the paper starts quite nicely but it is weakening toward the end. In particular, it is disappointing that not even a single result is presented. At least some overview results have to be included (which is also required according to the requirements for model experiment description papers in GMD), although a more in-depth discussion must of course be left to more specific papers.

In addition, some aspects of the model setup should be explained in more detail.

[Figure]

**Specific comments:**

The description of the model setup is not always precise. Several questions remain open illustrated below using the example of the model grids: Ìű

- According to the (too brief) section 4 WRF-Chem is run on a 0.25° x 0.4° lat-long grid. Was a rotated grid used for the WRF-Chem runs?

- Line 34 on page 6 and line 1 on page 7 indicate that WRF-Chem was run for a Lambert conformal grid at 25 km resolution ('A similar strategy [as for the meteorological driver for CMAQ] was used for WRF-Chem . . .'). This is contradictory to the statement above.

- According to Tab. 3 the horizontal grid width is approximately 25 km whereas the grid width of HIRLAM is approximately 22 km. However, in section 4 it is mentioned that all simulations were performed for the same lat-lon projection with 0.25° x 0.4°, with CMAQ being the only exception. Does this mean that MATCH has a different grid width than its meteorological driver? Please explain in more detail.

- Including an additional line with the name and the applied resolution of the meteorological driver in Table S1 might be helpful.

- Tab. 3 implies that the boundary conditions for the WRF-Chem simulation were derived from WRF simulations with 0.4°x 0.4° grid width. Is this true, or does 'WRF-0.44 simulation used by other EDT models' just mean that the WRF-Chem run and WRF-0.44 use the same ERA-Interim data for deriving the meteorological boundary conditions? If this was the case: Was meteorology nudging applied during the WRF-Chem run?

- Was nudging applied for RACMO and HIRLAM? This information could be added to Table 3.

Therefore, the setup of the models must be described in more detail.

Page 9, lines 21-22: Please add some more details and also remarks concerning the quality of these data.

Section 8.2: Please add some more details here (resolution, etc.)

Section 9: The 'Additional diagnostics' part looks just like copy paste from the modelling protocol. A table might be more useful. Eventually, this type of list might be moved to an appendix. Finally, although the reader can guess the meaning of all abbreviations e.g. O3_HL, it should be explained.

A few overview results including all contributing models (e.g. Taylor diagrams, box plots, for axamples see Solazzo et al., 2012 (doi:10.1016/j.atmosenv.2012.02.045) or Im et al., 2015, (http://dx.doi.org/10.1016/j.atmosenv.2014.09.042) or a table) should be added.

In their current form, the conclusions are more like a summary and outlook. But even the current outlook needs to be enhanced. Is there any concept concerning the further analysis of the results and future papers? Which detailed analysis is under work by the members of the consortium?

**Minor Points**

Further previous multi-model studies should also be mentioned in the introduction (at least multi-model studies with a minimum length of one year of simulation).

Page 5 line 9: Do the authors mean chemistry boundary conditions here?

Page 5: Is the WRF version (v3.3.1 according to Tab. 3) the same as described in Stegehius et al, 2015.?

Page 7: Section 6 is a bit meager. It should be either enhanced or incorporated into a section, which dedicated to all types of emissions.

[Figure]

Figure 2: The red dots as well as the blue dots are not well resolved in the figure, i.e most of the blue dots look more like a line.

[Figure]

---

## Author Comment (AC1) · 10 May 2017

Response to Anonymous Referee #1 for "EURODELTA-Trends, a multi-model experiment of air quality hindcast in Europe over 1990–2010" submitted to GMDD by Colette et al. 2016 as gmd-2016-309

Note: Referee comments are indicated in bold, answers are in regular font and changes highlighted in yellow in the revised manuscript

**General comment:**

The manuscript provides detailed information of the experiment set up. The experiments look reasonable. However, it is very difficult to understand the anticipated results based on the current information. As described in the journal website, results should be provided for model experiment description papers. (http://www.geoscientific-modeldevelopment.net/about/manuscript\_types.html#item4) *"Should include sufficient descriptions/figures of model results to give an overview of the project."* However, the manuscript does not provide any results. Without results, it is not easy for referees to provide further meaningful comments on the experiments.

In excluding model results from this experiment description paper, we followed the example of several recent articles of this category published in GMD for the Climate Model Intercomparison Projects (CMIPs). After further discussion with the topical editor, we understand that those were exceptional, and since the Eurodelta-Trend exercise is mature enough to include such sample results in the experiment description stage, we are pleased to address the referee's concern (also shared by referee #2) by adding a new section "Sample results". As explained in that section, we present some quickviews of the model ensemble. The evaluation of the ensemble and its ability to capture air pollution trends is one of the stated objectives of the experiment, it requires however a substantial analysis which will be the focus of forthcoming papers.

**Specific comments:**

**Line 12 in page 3: the "(" should be removed.**

Parenthesis removed

**Line 39-40 in page 3: the sentence is not completed.**

Sentence rephrased

**Line 32 in page 6: what does the "perfect" mean? I assume the authors refer to "better".**

In the regional climate modelling community (EuroCordex), boundary conditions originating from reanalyses are sometimes referred to as "perfect", in opposition when global climate model free-runs are used for historical periods. Since we are not really in that context here the word "reanalysis" is used now.

**Line 34-35 in page 8: What is the reason to rely on observation-based boundary for most experiments?**

Both approaches have pros and cons. The reason why observation-based boundary conditions were selected is the result of a consensus discussion in the modelling group concluding that the balance between pros and cons was in their favor. A choice had to be made to minimize the number of required simulations, even if we kept

sensitivity analyses with the model-based boundary conditions. The text has been revised slightly to explain that better.

**Line 18 in page 9: The ammonium trend based on NOx is not necessary true, as NH4 is also affected by NH3 emission which is expected to maintain at a constant level in the future, while NOx is expected to reduce.**

The reviewer is right to point out the uncertainties related to those trends. The results of the experiment are expected to address some of those points. Such uncertainty in the trends of the boundary conditions will certainly require further attention in the analysis. The sensitivity experiment using model-based boundary conditions will be used in that perspective. But, as far as observation based boundary conditions are concerned, we used the standard procedure used in EMEP modelling (Simpson et al., 2012). A sentence has been added at the end of the section to highlight those uncertainties.

**Line 34 in page 9: "The air concentration" should be "The air pollutant concentration". added**

Line 18 in page 12: The referee cannot agree this requirement based on two reasons. First, if users use the data, they should cite the corresponding publications. This should provide enough credits for the authors. Second, the findings of other studies may not be agreed by the authors. This may not be appropriate to include the authors' names on a paper that has findings against the authors' understanding. Therefore, this requirement should be modified such as "users of these data should cite this paper".

The present publication is a description of the experiment. We expect several forthcoming publications regarding model evaluation that need to acknowledge the work of contributing modelers. It should be stressed that the Eurodelta-Trends policy states that co-authorship must be "offered" to modelers, which keep their right to withdraw their names from an article they would not approve.

**Other updates**

The table of contribution of each modelling group to the various tiers has been updated to include new deliveries since the date of submission of the first manuscript. A problem has also been uncovered in WRF-Chem simulations, so that only an update of tier 1 is available for the updated contribution, while other tiers are now indicated as "planned".

---

## Author Comment (AC2) · 10 May 2017

Response to Anonymous Referee #2 for "EURODELTA-Trends, a multi-model experiment of air quality hindcast in Europe over 1990–2010" submitted to GMDD by Colette et al. 2016 as gmd-2016-309

Note: Referee comments are indicated in bold, answers are in regular font and changes highlighted in yellow in the revised manuscript

**General comment:**

**The paper describes a multi-model experiment including 12 one year long air quality simulations for Europe and two simulations with a length of twenty years. The volume of work presented in this model experiment description paper is really impressive and the paper should be published after some revision as a basis for the hopefully following numerous papers going more into the details.**

**In its currents form the paper starts quite nicely but it is weakening toward the end. In particular, it is disappointing that not even a single result is presented. At least some overview results have to be included (which is also required according to the requirements for model experiment description papers in GMD), although a more in depth discussion must of course be left to more specific papers.**

> In excluding model results from this experiment description paper, we followed the example of several recent articles of this category published in GMD for the Climate Model Intercomparison Projects (CMIPs). After further discussion with the topical editor, we understand that those were exceptional, and since the Eurodelta-Trend exercise is mature enough to include such sample results in the experiment description stage, we are pleased to address the referee's concern (also shared by referee #1) by adding a new section "Sample results". As explained in that section, we present some quickviews of the model ensemble. The evaluation of the ensemble and its ability to capture air pollution trends is one of the stated objectives of the experiment, it requires however a substantial analysis which will be the focus of forthcoming papers.

**Specific comments:**

**According to the (too brief) section 4 WRF-Chem is run on a 0.25 x 0.4 lat-long grid. Was a rotated grid used for the WRF-Chem runs?**

**Line 34 on page 6 and line 1 on page 7 indicate that WRF-Chem was run for a Lambert conformal grid at 25 km resolution ('A similar strategy [as for the meteorological driver for CMAQ] was used for WRF-Chem'). This is contradictory to the statement above.**

> WRF-Chem used the same lat-lon grid as all of the other models except CMAQB (the original text in Section 4 was correct). A revised explanation regarding the how meteorology was driven in the WRF-Chem runs has been added to Section 5:

"For WRF-Chem, an online model that simulates meteorology and chemistry simultaneously ("online"), the meteorology from the WRF-Eurocordex runs (Stegehuis et al 2015) was used as initial and lateral boundary conditions and for applying four-dimensional data assimilation (FDDA), with coefficients as described in Mar et al. 2016."

**According to Tab. 3 the horizontal grid width is approximately 25 km whereas the grid width of HIRLAM is approximately 22 km. However, in section 4 it is mentioned that all simulations were performed for the same lat-lon projection with 0.25 x 0.4, with CMAQ being the only exception. Does this mean that MATCH has a different grid width than its meteorological driver? Please explain in more detail.**

We have added such a clarification. We now explain that the original EURO4M meteorology was interpolated from the original 0.2 degree horizontal resolution on rotated lat-lon grid (corresponding to ca 22km resolution) to the grid of the CTM simulations.

**Including an additional line with the name and the applied resolution of the meteorological driver in Table S1 might be helpful.**

We have added this information in Table S1

**Tab. 3 implies that the boundary conditions for the WRF-Chem simulation were derived from WRF simulations with 0.4x 0.4 grid width. Is this true, or does 'WRF-0.44 simulation used by other EDT models' just mean that the WRF-Chem run and WRF-0.44 use the same ERA-Interim data for deriving the meteorological boundary conditions? If this was the case: Was meteorology nudging applied during the WRF-Chem run?**

The first option is true, WRF-0.44 data used by other models were used to force WRF-Chem at the boundaries, and for applying four-dimensional data assimilation (FDDA), with coefficients as described in Mar et al. 2016.

**Was nudging applied for RACMO and HIRLAM? This information could be added to Table 3.**

The HIRLAM reanalysis (EURO4M) uses data assimilation in 3 dimensions in the upper air (as explained in Table 3) and optimal interpolation for the surface fields. An initial analysis is conducted every 6 hours and three hourly forecast step saved and used by MATCH. We have added a clarification on this in the text. Thus, nudging was not used in producing the EURO4M data set.

The RACMO simulations are part of the EuroCordex ensemble documented in Jacob et al., (2013) and Kotlarski et al., (2014). We clarified in the text that it excludes nudging.

**Page 9, lines 21-22: Please add some more details and also remarks concerning the quality of these data.**

The following text has been added at the end of Section 7.1:

"The description of EMEP parameterization for sea spray and windblown dust can be found in Simpson et al. (2012). The accuracy of the model results for sea salt and mineral dust is regularly evaluated with available observations over Europe and documented in EMEP reports (www.emep.int). Model evaluation for mineral dust is limited due to the scarcity of dust in-situ measurements (see EMEP Status Report 1/2014), therefore also AOD/extinction measurements from satellite, Aeronet and Earlinet has recently been used for model evaluation within AeroCom aerocom.met.no)."

**Section 8.2: Please add some more details here (resolution, etc.)**

The following text has been added:

"The model uses a full tropospheric and stratospheric chemistry scheme (Lamarque et al., 2012) based on MOZART (Model for Ozone and Related chemical Tracers) version 4 (Emmons et al., 2010). CAM4-chem considers 56 vertical levels from the surface to about 40 km with 1.9° x 2.5° horizontal resolution. The simulation used in this analysis was performed in nudging the model to meteorological fields from the MERRA GEOS-5 (Modern Era Retrospective Analysis for Research and Application Goddard Earth Observing System Data Assimilation System Version 5) reanalysis provided by the Global Modelling and Assimilation Office (GMAO)."

**Section 9: The 'Addition al diagnostics' part looks just like copy paste from the modelling protocol. A table might be more useful. Eventually, this type of list might be moved to an appendix. Finally, although the reader can guess the meaning of all abbreviations e.g. O3_HL, it should be explained.**

Following the suggestion of the referee, we moved this section to a Table, also improving the language to avoid the style of modelling protocol as much as possible. All abbreviation should now be defined.

**A few overview results including all contributing models (e.g. Taylor diagrams, box plots, for examples see Solazzo et al., 2012 (doi:10.1016/j.atmosenv.2012.02.045) or Im et al., 2015, (http://dx.doi.org/10.1016/j.atmosenv.2014.09.042) or a table) should be added.**

See the answer to the general comment. We have added quickviews of the model ensemble but left out of the present article any comparison with observations. In following this approach, we opted for a similar strategy as the AQMEII model experiment description recently published as an ACP Tech. Note: http://www.atmos-chem-phys.net/17/1543/2017/

**In their current form, the conclusions are more like a summary and outlook. But even the current outlook needs to be enhanced. Is there any concept concerning the further analysis of the results and future papers? Which detailed analysis is under work by the members of the consortium?**

The ongoing analysis work is by nature an evolving object and it is difficult to write in an article to be published analysis plans that are still changing, but the main topics are, as stated in the conclusion:

- " … assess the capability of these state-of-the-art chemistry-transport models to reproduce the observed changes in the concentrations of the main pollutants …"
- " … assessment of the capability in reproducing the actual trends over the 21yr in the 1990-2010 period …"
- " … attribution of air quality trends to emission changes, to influx at the boundaries of the European domain, and to interannual meteorological variability … "
- "… serve for in depth analyses to scientific communities working on the impacts of air pollution on health, ecosystems or aerosol radiative forcing…"

**Minor Points**

**Further previous multi-model studies should also be mentioned in the introduction (at least multi-model studies with a minimum length of one year of simulation).**

We added a few references to the AQMEII project, in addition to the one-year multi-model publications from earlier phases of Eurodelta that were already cited. The following text has been added in the introduction:

"Over the recent past, several multi-model projects covering a time period of one year or less were undertaken such as the earlier phases of Eurodelta cited above but also the various phases of the AQMEII project (Galmarini et al., 2012;Galmarini et al., 2017;Rao et al., 2011;Im et al., 2015)."

**Page 5 line 9: Do the authors mean chemistry boundary conditions here?**

Yes, it has been corrected

**Page 5: Is the WRF version (v3.3.1 according to Tab. 3) the same as described in Stegehius et al, 2015.?**

Yes, the version number has been added in Section 5

**Page 7: Section 6 is a bit meager. It should be either enhanced or incorporated into a section, which dedicated to all types of emissions.**

We followed this suggestion by moving that part to a single "emission" section

**Figure 2: The red dots as well as the blue dots are not well resolved in the figure, i.e most of the blue dots look more like a line.**

The resolution of the figure has been improved

**Other updates**

The table of contribution of each modelling group to the various tiers has been updated to include new deliveries since the date of submission of the first manuscript. A problem has also been uncovered in WRF-Chem simulations, so that only an update of tier 1 is available for the updated contribution, while other tiers are now indicated as "planned".

---

## Editor Decision (ED1)

**Editor review for "EURODELTA-Trends, a multi-model experiment of air quality hindcast in Europe over 1990-2010" by Colette et al.**

Dear Authors,
   upon reading your revised manuscript I found the following issues which I like you to address in the final revised paper:

Major issues:

- I recommend to change the order of the Sections 2 and 3. As reader I expect first to read about the experimental design and afterwards something about the participating models.

- Sect. 2: Do I understand correctly, that this is a closed experiments, i.e., only the named 8 modeling teams can contribute? Anyhow, it would be good to enable other modelling groups to perform the same experiments. For this, the description is not detailed enough. Especially, the input data should be made available as well.

- Sect. 7 the title should be "Chemical Boundary Conditions". Or even "Lateral Chemical Boundary Conditions". Please clarify which Boundary Conditions are applied at the model top.

- Both types of chemical boundary conditions are provided as monthly averages. Thus certain chemical events, especially import from outside of the domain, e.g. dust or polluted air masses advected from the U.S., can not be modelled. Please discuss the consquences for the experiment results.

- Data availability: you just refer to the model output data. What about the input data? If other groups like to compare their own model to your results they need the same input data. Please make the input data also available for everyone. Especially, as you compared your own article to the ones of CMIP6, this is the large difference between your paper and the CMIP6 papers. Those papers have been written prior to the actual conductance of the experiments in order to provide all information about the experiments including access to all required input data.

Minor items:

- all citations included in the text should not be in brackets. e.g., p.3 L45, p. 4 l. 3, p. 8 l.5 / l. 6 / l.20, p. 9 l. 4, p. 10 l. 20 /l.37, etc.

- write acronyms always in the same way (e.g., EURODELTA in the title and Eurodelta in running text

- P. 4 l.15: Why is a link (footnote) and a citation provided for Polyphemus, but not for the other models?

- Sect. 4: why is a $0.25° \times 0.4°$ grid equivalent to $25km \times 25km$ ? I would have expected $25km \times 40km$.

- Please note, that Copernicus Office requires "last access dates" for all provided links. Best to provide them already in the final uploaded revised version.

- P.8 footnote 2: Move footnote indicator to front. / The link does not work.

- p.9 l.19: Why is this link not provided as a footnote?

- p.9 l.28: What do you mean by "mostly"?

- p. 12 l. 3: What do you mean by "available in 4"?

- p.12 l. 24-28: Any explanation for this outcome?

- Sect. 10 is more a "Summary and Outlook" section. Please consider renaming.

- Fig.1: In my printout the light-blue boxes are hardly visible, please change the colour.

- Fig.2: The black and the blue dots are hardly visible. You should not improve the resolution (as stated in your author reply) but the (line or points) width.

- Fig. 4 / Fig.5: increase the font size of the labels.

Best regards,
Astrid Kerkweg

---

## Author Response (AR2)

Response to Editor review for "EURODELTA-Trends, a multi-model experiment of air quality hindcast in Europe over 1990–2010" submitted to GMDD by Colette et al. 2016 as gmd-2016-309

Note: Editor comments are indicated in bold, answers are in regular font and changes highlighted in yellow in the revised manuscript

**Major issues:**

- **I recommend to change the order of the Sections 2 and 3. As reader I expect first to read about the experimental design and afterwards something about the participating models.**
  - Sections 2&3 have been swapped
- **Sect. 2: Do I understand correctly, that this is a closed experiments, i.e., only the named 8 modeling teams can contribute? Anyhow, it would be good to enable other modelling groups to perform the same experiments. For this, the description is not detailed enough. Especially, the input data should be made available as well.**
  - The experiment is not closed, and new models are still joining, although the analysis is well under way, and it there is a risk that newcomers might not be included in all forthcoming publications. The input data are available on a wiki through ESGF links that are now provided in the "data availability section".
- **Sect. 7 the title should be "Chemical Boundary Conditions". Or even "Lateral Chemical Boundary Conditions". Please clarify which Boundary Conditions are applied at the model top.**
  - The title of the section has been changed to "chemical boundary conditions" and a clarification has been added to explain that the same procedure is used at the model top.
- **Both types of chemical boundary conditions are provided as monthly averages. Thus certain chemical events, especially import from outside of the domain, e.g. dust or polluted air masses advected from the U.S., can not be modelled. Please discuss the consequences for the experiment results.**
  - A paragraph has been added to make this clearer also explaining that the impact on monthly means of such events are indeed taken into account.
- **Data availability: you just refer to the model output data. What about the input data? If other groups like to compare their own model to your results they need the same input data. Please make the input data also available for everyone. Especially, as you compared your own article to the ones of CMIP6, this is the large difference between your paper and the CMIP6 papers. Those papers have been written prior to the actual conductance of the experiments in order to provide all information about the experiments including access to all required input data.**
  - The link to the wiki and ESGF input data repository has been added

**Minor items:**

- **all citations included in the text should not be in brackets. e.g., p.3 L45, p. 4 l. 3, p. 8 l.5 / l. 6 / l.20, p. 9 l. 4, p. 10 l. 20 /l.37, etc.**
  - I am afraid that the Output Style available for Endnote on the Copernicus website is not handling this distinction properly, I hope this issue can be handled at the typesetting stage
- **write acronyms always in the same way (e.g., EURODELTA in the title and Eurodelta in running text**

- o   acronyms have been made homogeneous
- **P. 4 l.15: Why is a link (footnote) and a citation provided for Polyphemus, but not for the other models?**
  - o   The link to Polyphemus has been removed for consistency
- **Sect. 4: why is a 0.25 0.4  grid equivalent to 25kmx25km ? I would have expected 25km x 40km.**
  - o   At European latitudes, a 0.4degree longitude is closer to about 25km, a clarification has been added to the text.
- **Please note, that Copernicus requires "last access dates" for all provided links. Best to provide them already in the  final uploaded revised version.**
  - o   Access dates have been added
- **P.8 footnote 2: Move footnote indicator to front. / The link does not work.**
  - o   corrected
- **p.9 l.19: Why is this link not provided as a footnote?**
  - o   corrected
- **p.9 l.28: What do you mean by \mostly"?**
  - o   the sentence has been revised to explain that the agricultural sector largely dominates in European anthropogenic NH3 emissions
- **p. 12 l. 3: What do you mean by \available in 4"?**
  - o   The word "table" was missing
- **p.12 l. 24-28: Any explanation for this outcome?**
  - o   We are afraid that investigating the processes underlying such outcomes is typically the focus of forthcoming analysis papers
- **Sect. 10 is more a \Summary and Outlook" section. Please consider renaming.**
  - o   renamed
- **Fig.1: In my printout the light-blue boxes are hardly visible, please change the colour.**
  - o   changed
- **Fig.2: The black and the blue dots are hardly visible. You should not improve the resolution (as stated in your author reply) but the (line or points) width.**
  - o   Improving the resolution helped to see the red points, but the size of black and blue dots is now increased
- **Fig. 4 / Fig.5: increase the font size of the labels.**
  - o   The font size has been increased